# Phenotypes to remember: Evolutionary developmental memory capacity and robustness

**András Szilágyi**[1,2,3], **Péter Szabó**[1,4], **Mauro Santos**[1,5], **Eörs Szathmáry**[1,2,3]*

**1** Institute of Evolution, Centre for Ecological Research, Tihany, Hungary, **2** Department of Plant Systematics, Ecology and Theoretical Biology, Eötvös Loránd University, Budapest, Hungary, **3** Center for the Conceptual Foundations of Science, Parmenides Foundation, Pullach/Munich, Germany, **4** Department of Ecology, Institute for Biology, University of Veterinary Medicine Budapest, Budapest, Hungary, **5** Department de Genètica i de Microbiologia, Grup de Genòmica, Bioinformàtica i Biologia Evolutiva (GBBE), Universitat Autonòma de Barcelona, Barcelona, Spain

* szathmary.eors@gmail.com

**Data Availability Statement:** All relevant data are within the manuscript and its Supporting Information files. The source code of the implementation of the model is available on GitHub

## Abstract

There is increased awareness of the possibility of developmental memories resulting from evolutionary learning. Genetic regulatory and neural networks can be modelled by analogous formalism raising the important question of productive analogies in principles, processes and performance. We investigate the formation and persistence of various developmental memories of past phenotypes asking how the number of remembered past phenotypes scales with network size, to what extent memories stored form by Hebbian-like rules, and how robust these developmental "devo-engrams" are against networks perturbations (graceful degradation). The analogy between neural and genetic regulatory networks is not superficial in that it allows knowledge transfer between fields that used to be developed separately from each other. Known examples of spectacular phenotypic radiations could partly be accounted for in such terms.

## Author summary

The development of individual organisms from embryo to adult state is under the control of many genes. During development the initially active genes activate other genes, which in turn change the composition of regulatory elements. The behavior of genetic regulatory systems shows similarities to that of neural networks, of which the most remarkable one is developmental memory, the ability to quickly adapt to environments that have occurred in the past, occasionally several generations earlier. This is because each previously evolved developmental pathway leaves an "imprint" in the gene regulatory network. We investigated the properties of this system; the number of different developmental pathways that can be "memorized", how this number depends on the number of expressed genes, how fast the system can switch between these pathways, and its robustness against various disturbances affecting either the embryo state or the gene interaction networks.

(https://github.com/peterszabo77/developmental_memory_2020).

**Funding:** This work was supported by the National Research, Development and Innovation Office (NKFIH, https://nkfih.gov.hu) under OTKA grant numbers K124438 (A.S.), K119347 (E.S. and A.S.) and GINOP-2.3.2-15-2016-00057 (E.S., P.S. and A.S.) research grants; the ATTRACT Project EmLife (AS, ES and MS) (https://attract-eu.com); The Volkswagen Foundation (https://www.volkswagenstiftung.de) (initiative "Leben? –Ein neuer Blick der Naturwissenschaften auf die grundlegenden Prinzipien des Lebens", project "A unified model of recombination in life") (A.S., E.S. and M.S.). A.S. was supported by the Bolyai János Research Fellowship of the Hungarian Academy of Sciences (https://mta.hu/) and the ÚNKP-19-4 New National Excellence Program of the Ministry of Human Capacities. M.S. was supported by the research grant CGL2017-89160-P from the Ministerio de Economía, Industria y Competitividad, research grant 2017SGR 01379 from Generalitat de Catalunya, and the MTA Distinguished Guest Fellowship Programme in Hungary. All authors were supported by Templeton Foundation under grant number TWCF0268 ("Learning in evolution, evolution in learning"). The funders had no role in study design, data collection and analysis, decision to publish, or preparation of the manuscript.

**Competing interests:** The authors have declared that no competing interests exist.

Our results suggest that developmental memory may also provide the mechanism behind some rapid speciation processes.

## Introduction

Alan Turing, the father of machine learning, also formulated one of the most important mathematical models in developmental biology: the reaction-diffusion model for pattern generation [1]. This is striking because, although both gene regulatory networks [2–5] and associative neural networks [6–8] have extensive literature, only recently a conceptual analogy between evolutionary developmental processes and artificial neural network-based learning models has been articulated. [9–14]. Since development is the process whereby the phenotype is specified by the evolving genotype, late-evolved morphologies or functional capacities retain aspects of earlier stages ("memory") that were likely shaped by natural selection. These earlier stages might become reactivated if they are again useful in a different or a changing environment [15]. In this formulation evolutionary changes provide no novel structures that are non-homologous to an ancestral or existing one [16,17], but allow for recursion. For instance, mimetic color patterns of an extinct morph of the butterfly *Heliconius cydno*, presumably as a result of human disturbance, can be reconstructed from wild-caught butterflies [18], meaning that the morph could recur in nature if the former conditions reappear. Also surprising is the repeatability of evolution among closely related lineages [19,20]. An iconic textbook example is the extraordinary morphological convergence associated with adaptation to distinct ecological niches in cichlid fishes [21], with a large taxonomic diversity in the African Great Lakes Tanganyika (the oldest radiation, around 9–12 Myr ago with about 250 species), Malawi (less than 0.8 Myr ago and over 700 species) and Victoria (about 700 species evolved within the past 15,000 years) [22].

The idea that developmental processes can retain a memory of past selected phenotypes [13], together with the exceptional ability of genomes to find adaptive solutions that quickly converge upon remarkably similar states ("attractors" [23]) in closely related lineages, clearly suggests a non-linear genotype-phenotype mapping capable of producing multiple distinct phenotypes [13,24]. Non-linearity is also the hallmark of reaction-diffusion (Turing) and signaling systems involved in patterning processes [25], and developmental evolutionary biology (evo-devo) views the genotype-phenotype mapping as highly non-linear [26,27]. Furthermore, it might not be farfetched to think of some sort of developmental memory in the cichlid adaptive radiation. The explosive diversification in Lake Victoria was predated by an ancient admixture between two distantly related riverine lineages, one from the Upper Congo and one from the Upper Nile drainage [28]. Many phenotypic traits known to contribute to the adaptation of different ecological niches in the Lake Victoria radiation are also divergent between the riverine species [29,30]. Thus, when referring to the anatomical and morphological variation of Haplochromine cichlids, which are at the origin of the Lake Victoria radiation [28], Greenwood writes [30, p. 266]: "It is amongst the species of these various lacustrine flocks that one encounters the great range of anatomical, dental and morphological differentiation usually associated with the genus. The fluviatile species appear to be less diversified, but even here there is more diversity than is realized at first." If the high diversity in the Haplochromine cichlids of Lake Victoria is, to some extent, the result of re-evolved (similar) phenotypes in the ancestral fluviatile lineages, then the enduring question of why such an explosive diversification happened within a short time interval might have a simpler solution than previously thought. We aim here to sketch what the solution could be.

The genomic program for development operates primarily by the regulatory inputs and functional outputs of control genes that constitute network-like architectures [31], which are mathematically equivalent to artificial neural networks [10,11]. Although the insights of Vohradsky [10,11] and Watson et al. [13] shed light on an important analogy between neural and genetic regulatory networks, the conclusion of the theory of autoassociative networks cannot yet be readily extended to developmental systems. This is because of the different state space representations, as well as the nature of the task to be solved. Models of autoassociative networks tend to work with positive/negative state variables (inherited from ferromagnetic systems, but see [6]). In contrast to this, in ontogenetic systems the relevant space is that of nonnegative real numbers, corresponding to concentrations of different molecules, see e.g. [32]. Due to the nonlinear activation function features of models working with the above mentioned, alternative state representations can markedly differ. Another important consideration is that autoassociative networks (as their name indicates) solve the problem of the recovery of a particular state (attractor property). During ontogeny we require something more: not only should the adult stage be stable, but the system should reach this state from a particular embryo state (referred to as heteroassociative property). Moreover the system has to find transitions from different embryo states to corresponding, different adults states. In short, we require networks that solve problems of auto/heteroassociativity in one.

In the present investigation we extend the theory of gene regulatory networks to involve both auto- and heteroassociativity. We derive a heuristic formula for regulatory weights to obtain a functional system with the desired properties and we follow how Darwinian dynamics shapes the regulatory networks to acquire these properties. We compare the resulting regulatory matrices and analyze their robustness against different kinds of perturbation. Evolution of developmental pathways is interpreted within the context of ontogenetic dynamics.

## Methods

### Developmental model

Our model is a formal description of ontogenetic development operating primarily by the regulatory inputs and functional outputs of control genes. Consider an organism with $N$ genes. Its developmental state at time $t$, expressed by its gene product composition (e.g., proteins), can be represented by the vector $\mathbf{p}(t) = (p_1, p_2, \ldots, p_N)^{\mathrm{T}}$ with each element being the quantity of the product of a gene. These quantities are assumed to change due to protein decay and gene expression processes. Following [13], the ontogenetic dynamics of the developmental state can be described by the difference equation

$$p_i(t+1) = (1-\delta)p_i(t) + \tau f([\mathbf{Mp}(t)]_i), \tag{1}$$

where $\tau$ denotes the decay rate, $\tau$ denotes the maximal gene expression rate, $f(.)$ is the activation function, and the matrix $\mathbf{M}$ stands for the regulatory network. An $m_{ij}$ entry of the matrix gives the regulatory effect of the product of gene $j$ on the expression level of gene $i$; positive and negative elements imply activation and inhibition, respectively. The cumulative regulatory effects on any single gene $i$, i.e. the $i$th element of the product $\mathbf{Mp}$, determine the gene expressions via a sigmoid activation function modelled here as $f(x) = (1+\tanh(\omega x))/2$, where $\omega$ is the slope parameter.

From an ontogenetic viewpoint, the role of the gene regulatory network is to guide the individual along a developmental pathway from an initial embryonic state $\mathbf{p}(0) = \mathbf{e}$ to a specific adult state $\mathbf{p}(T) \rightarrow \mathbf{a}$. In real systems, an ensemble of different developmental pathways is desired, each responsible for achieving some environment-specific adult state from a particular

embryonic state. We used $T = 150$ iterations to reach the steady state, when the output does not change in time.

## Evolutionary model

In the evolutionary model we considered a population of $K$ individuals, with each member of the population represented by its regulatory matrix. All the interaction matrix elements were zero initially, representing an undeveloped regulation. Every individual shared the same environment, but the environment can change in time. We assumed $Q = 3$ number of different selective environments, each defining an embryonic state $\mathbf{e}^{(q)}$ and a corresponding adapted adult state $\mathbf{a}^{(q)}$. The selective environments alternated randomly; if the average fitness of the population approached the optimum ($\overline{w} > 0.95$ for at least 20 consecutive generations), or after 10000 generations, a new environment was chosen at random. In each generation the individuals underwent mutation, development and selection steps as follows.

Mutation: The mutation of the regulation network was implemented by adding a normally distributed random value, with zero mean and $\mu_W$ variance, to a randomly selected matrix element. Matrix elements were clipped into the range [−1,1].

Development: The equilibrium, adult state of each member of the population was obtained by iterating Eq (1).

Selection: The fitness of individual $k$ was expressed by a similarity index derived from the Euclidean distance between the actual adult state $\mathbf{p}(T)$ and the environment-specific optimal adult phenotype $\mathbf{a}^{(q)}$ as

$$w_k = 1 - \sqrt{\sum_{n=1}^{N} \left[ \frac{p_n(T) - a_n^{(q)}}{\tau/\delta} \right]^2} \tag{2}$$

Then the regulatory matrix of a randomly selected individual was replaced by that of the individual with the highest fitness (elitist selection). Although a stochastic Moran process [33] would be a more realistic selection scheme, for computational reasons simulations were performed using a relatively small population size ($K = 100$) that would result in too much genetic drift.

Embryonic and (optimally adapted) adult vectors: The number of genes was $N = 100$ with a low average expression level of $\sigma = 0.1$, where 40% of the expressed genes were common, 20% were partially common (appear in two pairs only), and 40% were unique in all the embryonic and all the adult vectors. Specifically, the expression sites of the employed state vectors were $\mathbf{e_1} = \{\underline{13}, \underline{19}, \underline{32}, \overline{36}, 39, 49, 55, \underline{72}, 81, \overline{87}\}$, $\mathbf{e_2} = \{\underline{13}, \underline{19}, 31, \underline{32}, \overline{40}, 60, 62, \underline{72}, \overline{87}, 100\}$, $\mathbf{e_3} = \{5, \underline{13}, \underline{19}, \underline{32}, \overline{36}, \overline{40}, 47, 67, \underline{72}, 94\}$, $\mathbf{a_1} = \{6, \overline{12}, 20, 24, \underline{46}, 65, \underline{84}, \overline{86}, 88, \underline{92}\}$, $\mathbf{a_2} = \{\underline{6}, 11, 28, \underline{46}, 79, \underline{84}, \overline{86}, \overline{91}, \underline{92}, 96\}$, $\mathbf{a_3} = \{\underline{6}, \overline{12}, \underline{46}, 56, 61, 66, 80, \underline{84}, \overline{91}, \underline{92}\}$;

where underlines and overlines denote the common and partially common elements, respectively. E.g. gene 13 is expressed in all embryo states (common element), gene 12 is expressed in two adult states (partially common element). The initial state was always a perturbed embryonic state. The perturbation was performed, similar to the mutations, by adding a normally distributed random value, with zero mean and $\mu_e$ variance, to a randomly selected element of the environment-specific embryo vector. Vector elements were clipped into the range [0,$\tau/\delta$].

## Perturbation analysis

To investigate the robustness of the resulting gene regulatory networks we evaluated their performance against three different kind of perturbations. The embryo states were perturbed by

flipping the vector elements from low to high, or vice versa, with the given probability. The interaction matrices were perturbed by either adding random values to all matrix elements, drawn from a normal distribution with the given standard deviation, or by nullifying a proportion of the elements. Note that, in the evolutionary algorithm we perturbed only single elements of the embryonic states. In contrast, in the analytical matrix construction we perturbed all elements of the embryonic vectors to incorporate the accumulating effects of many consecutive perturbations on the interaction matrix.

## Results

To perform the developmental task, the network must guarantee that (i) each adult state is a stable equilibrium point of the dynamics (stability condition), and (ii) each embryonic state is within the basin of attraction of its corresponding adult state (attraction condition); these two conditions correspond to the auto- and heteroassociative properties in a neural network [34]. Note that this is a more difficult task than a simple pattern recovery problem, which is known to be achievable by a neural network with the standard Hebbian learning rule that fulfils only the stability condition [7]. Not only must all the adult states have a basin of attraction, but these basins must include the corresponding embryonic states.

We found that the task-optimized structure of the regulatory network can be inferred from the embryo-adult state vector pairs in the form of an interaction matrix **M** (Fig 1). Consider the simplest case with one embryo-adult pair (i.e. one developmental pathway). Depending on whether a gene is expressed in the adult state or not, all the other expressed gene products, in either the embryonic or the adult state, must enhance or block its expression, respectively. This would provide, on the one hand, stability for the adult state and, on the other hand, attraction from the embryonic state. Note, however, that if a gene is expressed in neither the embryonic nor the adult state, then its regulatory effect is irrelevant, therefore the corresponding matrix elements are undetermined. In summary, an $m_{ij}$ element of the regulatory matrix **M** should be positive or negative, depending on whether the $i$th gene is expressed in the adult state or not, except when the $j$th gene is expressed in neither the embryonic nor the adult state. The above line of thought can be generalized for arbitrary $Q$ number of embryo-adult state pairs. Denoting the zero-one normalized embryonic and adult state vectors by **e** and **a**, such a matrix can be obtained by averaging two dyadic products for all developmental pathways as

$$\mathbf{M} = \frac{1}{2Q} \sum_{q=1}^{Q} (2\hat{\mathbf{a}}^{(q)} - 1) \circ \hat{\mathbf{a}}^{(q)} + (2\hat{\mathbf{a}}^{(q)} - 1) \circ \hat{\mathbf{e}}^{(q)}, \tag{3}$$

where $Q$ stands for the number of embryo-adult state pairs and $q$ denotes the different pairs. The first and second dyadic products are responsible for the stability and attraction conditions, respectively. Note the similarity of our treatment and Kurikava and Kaneko 2012 and 2013 approach [35,36]. Within each dyadic product the right term determines whether an entry is relevant from the viewpoint of the state vector, whereas the left dyadic term determines its sign. The resulting matrix contains positive values, negative values and zeros for activator, inhibitory and undetermined elements, respectively. Notice that the developmental pathways can be in conflict with each other as to whether a gene should be up- or downregulated by another gene. It is instructive to compare this formula with the standard Hebbian learning rule **H** = **a**∘**a** for $a_i \in \{-1,+1\}$. Its modification for $a_i \in \{0,+1\}$ vectors that preserves that stability condition is **H** = (2**a**−1)∘**a**, which is identical to the first term in Eq (3), c.f. Table 1, and see [6]. (This is the standard procedure in ANNs with unsigned states as it converts the training patterns to signed values even though the state vector is unsigned. Natural selection will not operate like this–we are just showing how to hand-construct a solution that works.)

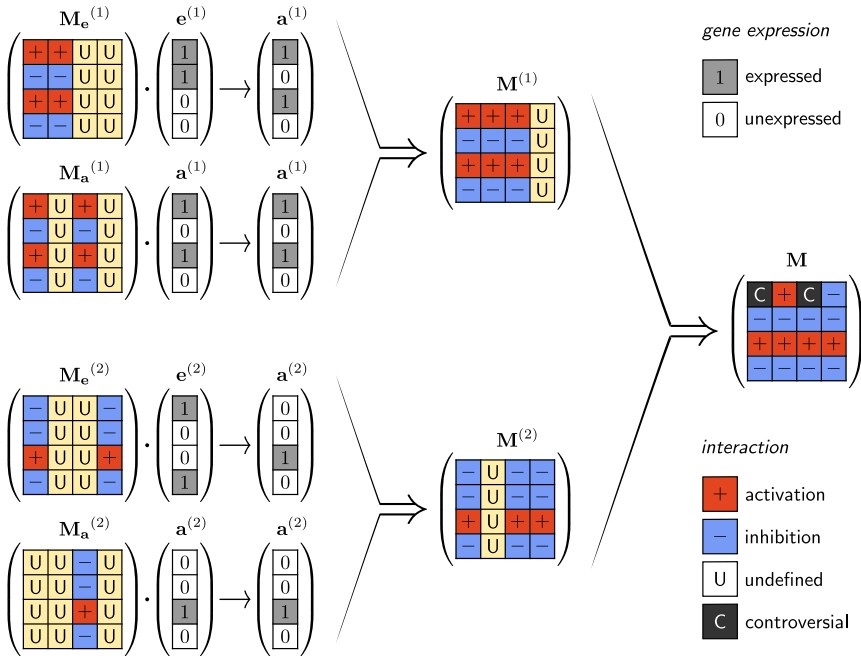

**Fig 1. Illustration of the construction rules of interaction matrices based on theoretical considerations on the optimal pairwise interaction types between genes. $\mathbf{e}^{(1)}$ and $\mathbf{a}^{(1)}$ are the first embryo-adult pair, $\mathbf{e}^{(2)}$ and $\mathbf{a}^{(2)}$.** the second pair. Depending on the combination of gene expressions $e_i^{(n)}$ and $a_i^{(n)}$ in an embryo-adult vector pair ($n = 1,2$), an $m_{ij}$ element of the interaction matrix can be positive ($'+'$, activation), negative ($'-'$, inhibition), or undefined ($'U'$). To ensure correct development ($\mathbf{M}_{\mathbf{e}}^{(n)}\mathbf{e}^{(n)} \rightarrow \mathbf{a}^{(n)}$) the $\mathbf{M}_{\mathbf{e}}^{(n)}$ matrices must have the structure indicated in the figure. (If $e_j^{(n)} = 1$ and $a_i^{(n)} = 1$, then $m_{ij}^{(n)} = '+'$; if $e_j^{(n)} = 1$ and $a_i^{(n)} = 0$, then $m_{ij}^{(n)} = '-'$; if $e_j^{(n)} = 0$, then $m_{ij}^{(n)} = 'U'$; irrespective of the value of $a_i^{(n)}$.) A similar argument holds for the stability criteria ($\mathbf{M}_{\mathbf{a}}^{(n)}\mathbf{a}^{(n)} \rightarrow \mathbf{a}^{(n)}$) and results in the $\mathbf{M}_{\mathbf{a}}^{(n)}$ matrices. By combining $\mathbf{M}_{\mathbf{e}}^{(n)}$ and $\mathbf{M}_{\mathbf{a}}^{(n)}$ the resulting $\mathbf{M}^{(n)}$ fulfills both the attractivity and stability criteria. The combination rules are the following: $(+,+)\rightarrow+$; $(-,-)\rightarrow-$; $(\pm,U)\rightarrow\pm$; and $(\pm,\mp)\rightarrow C$, which can be done practically by taking the element-wise average of the two matrices. The ultimate combination of all $\mathbf{M}^{(n)}$s results in a matrix that fulfills the attraction and stability criteria for all different embryo-adult pairs.

We investigated the parameter dependence of the analytic model. As for the regulatory matrix we used a slightly modified version of Eq (3). Treves [8] claims that the interaction terms should be modified by the average expression $\sigma$, i.e. the proportion of expressed genes. This is because if a larger proportion of genes is expressed, then proportionally smaller interaction strengths are needed for the same regulatory effect on any single gene. Incorporating this

**Table 1. Comparison of the resulting analytic interaction matrices for an autoassociative task with the two representations.**

|  | {-1,+1} representation | {0,+1} representation |
|---|---|---|
| learning rule | $H_{ij} = \sum_q a_i^{(q)} a_j^{(q)}$ (Hebb$-$rule) | $M_{ij} = \sum_q (2a_i^{(q)} - 1) a_j^{(q)}$ |
| symmetry | $H_{ij} = H$ | non-symmetric |
| neutralities in weight matrix | no neutral elements | can be neutral elements ("opposite to" zero vector elements) |
| main diagonal | always positive (if allowed) | can be negative or positive |
| structure | has a unique structure | many different realizations |

consideration into Eq (3) gives

$$\mathbf{M} = \frac{1}{2Q} \sum_{q=1}^{Q} (2\hat{\mathbf{a}}^{(q)} - 1) \circ (\hat{\mathbf{a}}^{(q)} - \sigma + \hat{\mathbf{e}}^{(q)} - \sigma) \qquad (4)$$

The performance of a regulatory network constructed by the above rule changes with the number of developmental pathways and gene expression levels (Fig 2). With increasing number of embryo-adult pairs, the accumulating conflicts between them inevitably corrupt the regulatory ability of the network; some adult states will be unreachable from their embryonic states. Nevertheless, the network is able to tolerate a fair number of conflicts, related to its structural stability. Since conflicts can occur only between non-orthogonal state vectors, the performance of the network also depends on the amount of overlap in the expression patterns of states belonging to different pairs. This highlights the importance of the proportion of expressed genes; i.e., the sparseness of the state vectors. If these vectors are very sparse, then they are likely to be orthogonal, therefore the number of learnable embryo-adult pairs does not reduce. The number of learnable pairs is a decreasing function of sparseness, because the numerous non-orthogonal state vectors and the largely different gene expressions in the adult states lead to several conflicts among them. Regarding the effect of system size on functionality, the results are in line with the expectations; the higher the number of genes, the higher is the number of "error free" developmental pathways (Fig 3).

We have also analyzed the "memory capacity", i.e. the number of learnable developmental pathways as a function of the system size ($N$). There are several studies on the memory capacity of Hopfield networks [e.g. 37,38], but these results are not applicable to our hetero/autoassociative system, and the capacity definitions themselves are valid only in the $N \rightarrow \infty$ limit, while our systems are relatively small ($N \leq 500$). Therefore, we define the memory capacity of a given system as the average number of developmental pathways which can be reconstructed at $r = 0.95$ performance (denoted by $Q_{0.95}$, c.f. Fig 2C). Fig 4 shows that $Q_{0.95}$ depends linearly on $N$: $Q_{0.95} = \vartheta(\sigma) \cdot N + b(\sigma)$.

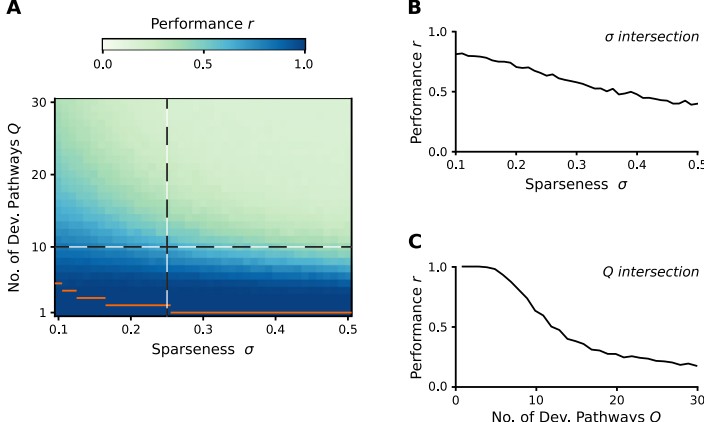

**Fig 2. Performance of the analytic developmental networks.** We assumed different sparseness (proportion on non-zero entries in the state vectors) values and different number of embryo-adult pairs. Embryo and desired adult vectors were generated by independently setting each vector element to high or low randomly according to the sparseness value. The performance was measured by the averaged (over 400 realizations) Pearson correlation(s) between the desired and the experienced adult state(s) for all developmental pathways (panel A). Panels B and C show a more detailed view for the two cross-sections of the parameter space (indicated by dashed lines in panel A). Orange horizontal lines show the maximum number of orthogonal state vectors for the given sparseness values. Parameters: $N = 100$, $\delta = 0.2$, $\tau = 1$, $\omega = 25$).

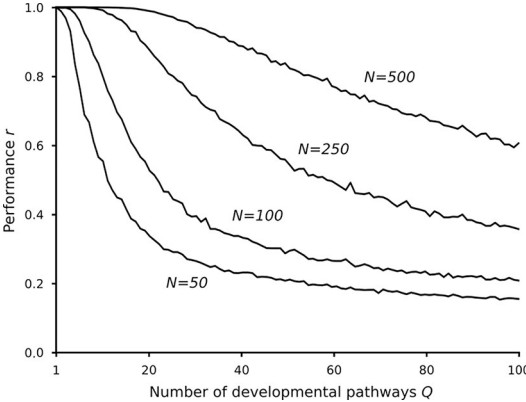

**Fig 3. Performance of the analytic developmental networks with different number of genes and developmental pathways.** Relevant parameters as in Fig 2 and $\sigma$ = 0.1.

Since decreasing sparseness increases the probability of controversial interactions (cf.. 1) $\vartheta$ ($\sigma$) is a decreasing function of $\sigma$, in line with the expectations. The $\vartheta$ values, which are in the range [0.0588, 0.0236] depending on $\sigma$, can be treated as a proportionality factor of memory capacity. These are significantly lower values than the proportionality value of 0.138 in the widespread used linear formula for autoassociative networks, see e.g. [39]. Notice, however, that these values should be compared with caution on one hand due to the different interpretation of memory capacity, and on the other hand owing to the "double task" nature (hetero- and autoassociative tasks) of our system.

A key question is whether a functional network is attainable by Darwinian selection via a series of mutation-selection steps. In our evolutionary model we used a more realistic Darwinian dynamics than the solitary stochastic hill climbing [13]. From the viewpoint of the theory of artificial neural networks this process can be regarded as a Darwinian dynamics-driven learning process. The evolutionary algorithm yields interaction matrices that contain positive and negative values where the heuristic formulation predicts them (Fig 5). While the individual interaction matrices vary, their average is in line with the heuristically derived matrix. The values are arranged into a characteristic structure; positive and negative entries form horizontal stripes, intermitted with vertical stripes of near-zero values (c.f. Fig 1). Those genes have the

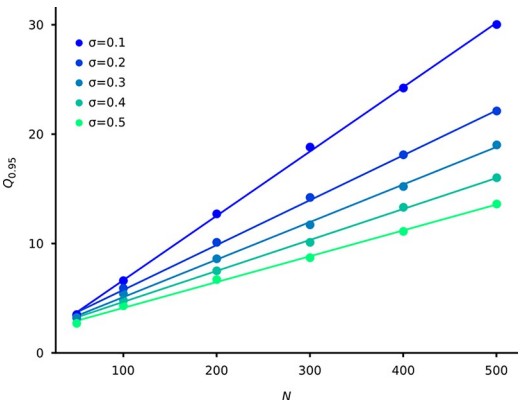

**Fig 4. The memory capacity ($Q_{0.95}$) as a function of the number of neurons ($N$) at different sparseness ($\sigma$).** Similar to Fig 2, each point is an average of 400 realizations. The lines are the linear regression fits ($R^2 > 0.99$ in all cases).

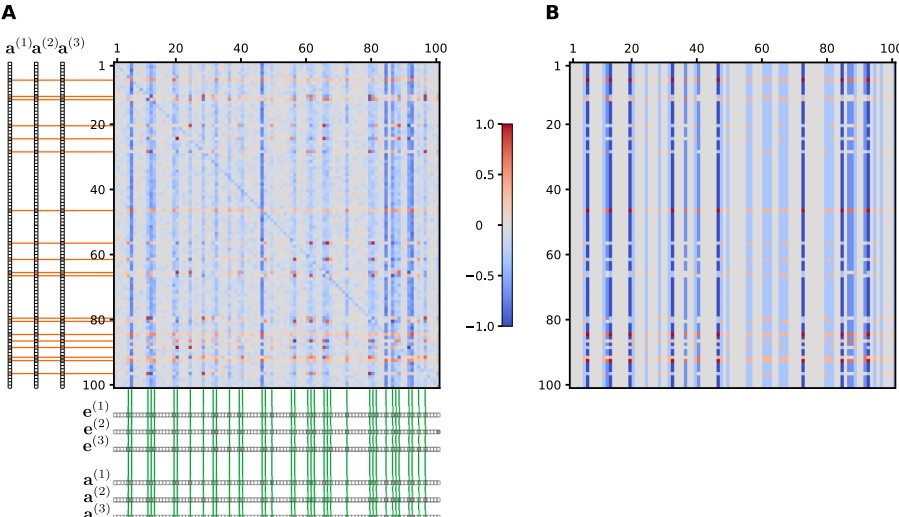

**Fig 5. Structure of the interaction matrix obtained with the evolutionary algorithm as compared to the analytically derived one for three developmental pathways.** (A) The evolutionary interaction matrix was obtained by averaging the output of 300 independent runs of the evolutionary algorithm. The three applied environment-specific embryonic and adult state vectors are shown along the sides. Orange guidelines highlight those rows where the corresponding genes are expressed in at least one adult state, whereas green guidelines highlight those columns where at least one gene is expressed in any of the embryonic or adult states. (B) The theoretically predicted interaction matrix was constructed from the embryonic and adult state vectors using Eq (3). Parameters as in Fig 2 and $Q = 3$, $K = 100$, $\mu_W = 0.05$, $\mu_e = 0.1$, $\sigma = 0.1$.

largest effect on the developmental process, which are expressed in any embryonic or adult states (c.f. marked columns in Fig 5). Depending on whether the affected gene is expressed in any of the adult states, they have a strong positive or negative effect (c.f. marked rows in Fig 5). The rest of the genes drift freely in individual realizations due to a lack of selective pressure. Consequently, the average values in these positions are approximately zero (c.f. grey columns in left panel of Fig 5). The corresponding values in the analytic treatment (undefined elements) are zero by definition. The only major difference from the heuristic matrix is that the main diagonal elements of the evolutionary matrix are mainly negative, which means that the expression of every gene is under negative feedback by its own inhibitory product. A possible explanation is that without a strong negative feedback a gene could be easily overexpressed due to the perturbations of the interaction elements. This is more probable if the sparseness of the expression vectors is low, as it was in our case. This picture is likely to change with hierarchical developmental regulation, the evolution of which takes longer time and should be investigated in the future.

A detailed view of the evolutionary process is shown in Fig 6. During the early generations, where the gene regulation is undeveloped, it takes many generations (i.e., mutation-selection steps) to approach the environment-specific optimum. In addition, selection for one environment can have adverse effects on performance in another environment if the basin of attraction of the actually selected adult state engulfs the neighborhood of the embryonic states of other adult states. But those interactions which are not beneficial in any of the environments are eliminated. Deleterious mutations may arise any time also in a well-functioning system, but selection eliminates them over the timescale of a few environmental changes.

A developmental process must be sufficiently robust against stochastic perturbations of both the embryonic state and the gene interaction matrix [5,12]. It requires that the neighborhood (according to a given metric) of the embryonic states must also be in the basin of their

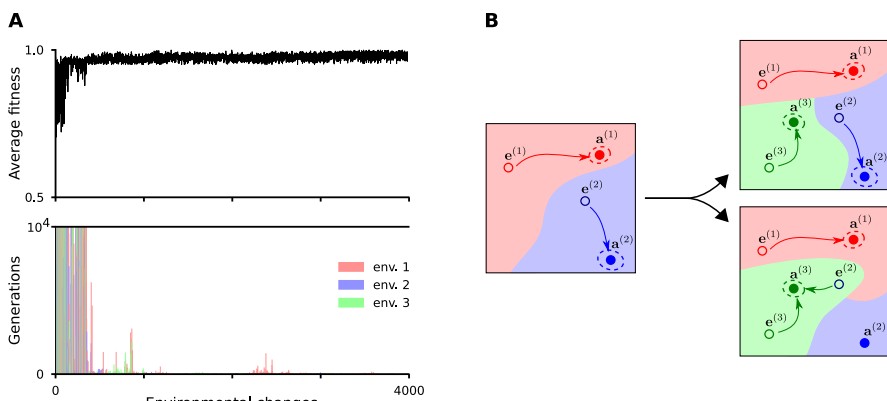

**Fig 6. Learning of three different developmental pathways in the evolutionary model.** (A) Average fitness and the mutation-selection steps needed to achieve a well-functioning developmental network during random environmental changes. The three environments are denoted by red, green and blue. Parameters as in Fig 5. (B) Schematic illustration of the changes in the state-space dynamics during the evolutionary process with three developmental pathways (indicated by red, blue and green colors). The panels show the basins of attraction of an initial, random regulation system with two embryo-adult pairs (left), a well-functioning one (top right) and a bad one, where the basins of attraction of the adult states (filled dots encircled by dotted lines indicating variation around the target phenotypes) include not only their corresponding embryo states (bottom right).

corresponding adult states. Therefore, some inputs of variation should produce little or no phenotypic variation at all, a phenomenon that has received a lot of attention under the labels of canalization, robustness or buffering [31,32,40–42]. The recovery performance of the network changes with increasing amount of perturbations (Fig 7). The system is very robust against perturbations regarding the embryonic state, it is moderately robust against additive perturbations and has limited robustness against eliminated interactions regarding the interaction network. The sharp difference between the robustness of evolutionary and analytic models in the latter case is the consequence of the peculiarities of the evolutionary process. In the evolutionary matrix the regulatory weights are less evenly distributed as compared to the analytic one, due to the stochastic nature of the evolutionary process. It makes the system more sensitive to eliminating perturbations. Resilience understandably decreases with the number of developmental pathways in all cases, but conforming to "graceful degradation" in artificial neural networks; i.e., performance first decreases mildly and drops fast only beyond a critical strength of perturbation [7]. To sum up, variation is apportioned into discontinuous (basins of attraction) and continuous (small perturbations around the target) phenotypes (Fig 6B). Evo-devo mainly focuses on the first kind of variation whereas standard evolutionary genetics focuses on the second [26,43].

## Discussion

Treating gene regulatory networks as formally analogous to artificial neural networks [10,11] allows translating the well-known dynamics of the latter [44] to model genomic programs for development. There is widespread natural variation in morphogenic pathways [45], and the developmental memory of past selected phenotypes [13] is akin to the memory capacity of neural networks. This developmental memory allows populations to re-evolve phenotypes much faster than it would be possible if they had to evolve de novo. Previous speculation on the effect of the heat-shock protein Hsp90 as a capacitor for releasing hidden morphogenetic variation that could allow fast morphological radiations [45] has been criticized on the grounds that the function of Hsp90 is to prevent morphological aberrations. Furthermore,

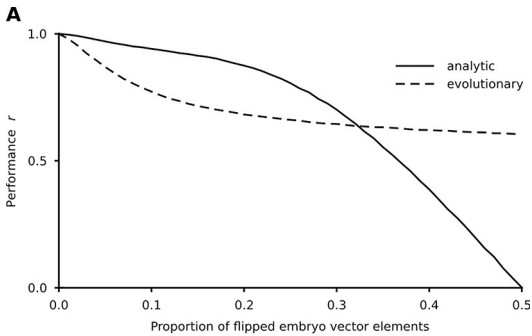

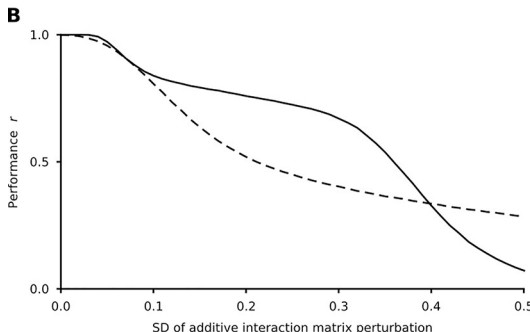

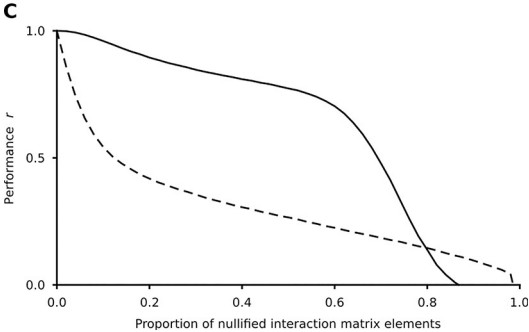

**Fig 7. Robustness of the developmental dynamics against perturbations.** The interaction matrices were constructed from the given number of embryo-adult vector pairs according to Eq (4). The performance was expressed by the Pearson correlation(s) between the desired and the experienced adult state(s) for all developmental pathways after $T = 150$ iterations averaged over 300 matrices and 100 perturbations for each parameter combination. (A) Performance against the proportion of the flipped embryonic vector elements. (B). Performance against the standard deviation ($SD$) of the perturbation of the interaction matrix. All elements of the matrix were perturbed additively by an $N(0,SD)$ random number. (C) Performance against the proportion of nullified elements of the interaction matrix. Each element of the interaction matrix was set to zero with the given probability. Relevant parameters are as in Fig 5.

some sense of purposive evolution, fully incompatible with the lack of foresight of natural selection, lays behind this sort of interpretations [46].

These criticisms do not apply here because in our developmental model past selected states can recur in the population if they appear useful again in a different environment or body context. As any theoretical model, ours obviously has inherent limitations and highly simplifies the representation of biological systems. However, to the extent that it captures sufficient conditions to generate the phenomenon of morphological radiations, more complex explanations are not required. Thus, the assumption that structural novelties (or "key innovations") are associated with adaptive radiations into new ecological niches (e.g. [47, p. 159]) might be

unwarranted. There is a noteworthy implication in the foregoing consideration for the understanding of atavism. Crocodilian teeth can grow in mutant birds, which suggests the reactivation of the associated developmental machinery [48], that required the resurrection [49] of a key aspect of regulation. The same neurons participate in the storage of different engrams in neural networks. The same holds for the storage of devo-engrams in genetic regulatory networks. Resurrection leading to atavism requires only limited reactivation of a few connections in a network that is maintained by the current selective forces. An exciting question is how evo-devo learning can generalize from the "training set" (previously selected target phenotypes) to novel ones [13,50]. The prediction described by these authors is that generalization potential works within a set that can be characterized by the same formal grammar.

While the theory of neural networks can (and does) infer the same conclusions based on different representations, in the case of modelling real biological situations the adequacy of the representation can be crucial (the same holds for neuronal networks). Our results show that a linear change to the representation has profound impact on the essential features of the system. While in the customary (neural) {-1,+1} representation there are no neutral elements in the interaction matrix, the biologically adequate {0,+1} representation of genetic regulatory networks allows for the free choice of interaction elements being opposite to "0". This feature turns out to increase the robustness of the system against the disturbance of interaction coefficients only if the system is very sparse, which guarantees the commonly zero elements in the embryo and adult states. Another feature of our representation is the large number of different interaction matrices entailing the same developmental process, thus evolution "from scratch" does not face so many constraints. In other words, starting with a single ancestor an extraordinarily rapid morphological diversification could be attained, which is the hallmark of adaptive radiations.

## Acknowledgments

The authors thank Ferenc Jordán and Harold. P. de Vladar for comments on an earlier draft.

## Author Contributions

**Conceptualization:** András Szilágyi, Eörs Szathmáry.

**Data curation:** András Szilágyi, Péter Szabó.

**Formal analysis:** András Szilágyi, Péter Szabó.

**Funding acquisition:** Eörs Szathmáry.

**Investigation:** András Szilágyi, Péter Szabó, Eörs Szathmáry.

**Methodology:** András Szilágyi, Péter Szabó.

**Software:** András Szilágyi, Péter Szabó.

**Supervision:** Eörs Szathmáry.

**Validation:** András Szilágyi, Péter Szabó.

**Visualization:** András Szilágyi, Péter Szabó.

**Writing – original draft:** András Szilágyi, Péter Szabó, Mauro Santos, Eörs Szathmáry.

**Writing – review & editing:** András Szilágyi, Péter Szabó, Mauro Santos, Eörs Szathmáry.

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
