## [Decision Letter · Decision Letter 0]

7 Sep 2020

Dear Professor Szathmáry,

Thank you very much for submitting your manuscript "Phenotypes to remember: Evolutionary developmental memory capacity and robustness" for consideration at PLOS Computational Biology.

As with all papers reviewed by the journal, your manuscript was reviewed by members of the editorial board and by several independent reviewers. In light of the reviews (below this email), we would like to invite the resubmission of a significantly-revised version that takes into account the reviewers' comments.

Please note that apart from Reviewer #1's narrative comments below he also provided an annotated copy of your manuscript (see attachment). Reviewer #2 asks you to cite earlier work by Kauffman, Kaneko, and Newman. I would elaborate on this by noting that Kauffman and Kaneko have each presented gene regulatory network systems, which like the one in the present manuscript, generates systems of hierarchically nested attractors that serve as models of development. Newman, in contrast, has presented not a dynamical network treatment, but a perspective in which inherent physical properties of developmental systems provide a basis for understanding recurrent features in evolution. This is also a precedent for the present work. Newman has also recently presented a critical view of the sufficiency of dynamical systems models for evolutionary simulations (doi: 10.1016/j.jtbi.2019.110031). The authors may or may not wish to address this critique.      

We cannot make any decision about publication until we have seen the revised manuscript and your response to the reviewers' comments. Your revised manuscript is also likely to be sent to reviewers for further evaluation.

Sincerely,

Stuart A Newman

Guest Editor

PLOS Computational Biology

Natalia Komarova

Deputy Editor

PLOS Computational Biology

Reviewer's Responses to Questions

**Comments to the Authors:**

Reviewer #1: This is a very clear and well-developed paper. It develops work on the functional equivalence of learning and evolution - specifically, in the context of GRNs and correlation learning, and the ability to evolve GRNs that store multiple phenotypic memories. The work is carried out carefully and presented clearly - I see no technical problems. The results are clear and not overstated. The finding - that the functional equivalence of evolved GRNs and correlation learning models can be deepened and extended - is significant. Previous work in this area used simplifying assumptions, and assessing whether the effects depend on these assumptions is important. The work also extends the scope of evolved behaviours from autoassociative to heteroassociative - and this is a significant extension. The following comments are therefore minor/suggestions for improvements.

1. I like an introduction to be clear about the aims and claims. As it is, the intro is excellent at setting the scene but it could be more clear about the hypotheses that will be tested or the questions that will be answered. This has a knock-on effect later in the paper.

2. It seems like the aim is primarily to figure out whether/how the simplifying assumptions of prior work, compared to a more realistic modelling set-up for the GRN, matters to the idea of evolution learning multiple developmental memories. It turns out, by the end of the paper, that the analogy remains sound, but there are some interesting differences along the way. The results also show that the task can be extended to heteroassociative tasks which hasn’t been done before. Clarity on the balance of these two findings could improve the paper. Specifically,

- If the aim of the paper was really to detail the impact of these different implementational assumptions, it would have been valuable to compare the two models on the same task. I don’t think that the non-negative state models work on the tasks of earlier work. And you’ve showed that the non-negative state models do work on your task. But we don’t have a side-by-side comparison of both models on both tasks.

- Likewise, you suggest that the non-negative state model results in more neutrality and that this increases robustness (compared to the model with signed states). I don’t think its clear that the signed state model has a problem with robustness despite having no neutrals. And we don’t have a side-by-side comparison.

- However, if the claim of the paper is that these differences don’t matter much – ie that the analogy is basically sound, and the task space can be extended, then a side-by-side comparison is not needed.

- I think that the implementational details do matter to some tasks – and your paper is very useful in helping us think that through more carefully. And the implementational details sometimes don’t matter - and you’ve shown a case where they don’t. So, both points are worth making. Yet, I think the balance in the writing at present is a little bit towards the former, whereas the results better support the latter. So, perhaps the paper could be improved if you stated your aims/claims/questions/hypotheses more clearly at the beginning in the intro.

3. A technical point relating to contrast of signed and unsigned models. You say (Table 1) that signed states will result in Hebbian learning and symmetric interaction terms (as observed in prior work). But the evolved interactions in that work are symmetric because there is no reason for them to be asymmetric – not because they have to be symmetric or because Hebb’s rule is applied. Hebb’s rule isn’t applied, of course – and they aren’t forced to be symmetric - but when there is no reason to be asymmetric, symmetric Hebbian changes are observed. When evolving weights to a single pattern with unsigned states, the weights are not symmetric and not Hebbian – as you say. But I happen to know that in other conditions (e.g. when connections are costly) asymmetric interactions evolve even when the states are signed. So, question: Is there a reason for selection to create asymmetric weights in the heteroassociative task, does it require asymmetric weights (your Fig 1 suggests yes), and would asymmetric weights then evolve even in the signed state model? If we don’t know the answers to these questions, then Im not sure that all the rows of your Table 1 follow from the choice of signed/unsigned state models. As per 2 above, Im not sure that such questions/issues regarding the comparison of the modelling choices are the main contribution of this paper – in which case this doesn’t matter so much.

4. Since capacity and robustness are in the title….

a. There is early (1980s) and recent work on Hop Net capacity (e.g. Folli, V., Leonetti, M., & Ruocco, G. (2017). “On the maximum storage capacity of the Hopfield model”. Frontiers in computational neuroscience, 10, 144. and refs therein)

b. Fig 6 top and middle supports your claims nicely wrt robustness under large perturbations. Im not sure that the bottom fig does though – If Im reading it right. It does have a larger tail in the limit of large perturbations, but the initial drop off is large – I don’t see how this supports the claim that neutrality increases robustness.

5. some other (optional) suggestions and thoughts for consideration are provided in the commented pdf

In sum, this paper makes a very strong contribution to the topic, and very nicely written and easy to read - with strong technical analysis. I hope these suggestions above are helpful.

Reviewer #2: (attached)

**Have all data underlying the figures and results presented in the manuscript been provided?**

Reviewer #1: None

Reviewer #2: Yes

PLOS authors have the option to publish the peer review history of their article (what does this mean?). If published, this will include your full peer review and any attached files.

Reviewer #1: **Yes: **Richard A. Watson

Reviewer #2: No
---

## [Editor Report · Decision Letter 1]

6 Oct 2020

Dear Professor Szathmáry,

We are pleased to inform you that your manuscript 'Phenotypes to remember: Evolutionary developmental memory capacity and robustness' has been provisionally accepted for publication in PLOS Computational Biology.

Best regards,

Stuart A Newman

Guest Editor

PLOS Computational Biology

Natalia Komarova

Deputy Editor

PLOS Computational Biology

---

## [Editor Report · Acceptance letter]

28 Oct 2020

PCOMPBIOL-D-20-00996R1 

Phenotypes to remember: Evolutionary developmental memory capacity and robustness

Dear Dr Szathmáry,

I am pleased to inform you that your manuscript has been formally accepted for publication in PLOS Computational Biology. Your manuscript is now with our production department and you will be notified of the publication date in due course.

With kind regards,

Melanie Wincott
